# Pathophysiological Links Between Myocardial Infarction and Anxiety Disorder, Major Depressive Disorder, Bipolar Disorder and Schizophrenia

**DOI:** 10.3390/biology14040336

**Published:** 2025-03-25

**Authors:** Leong Tung Ong, Ching-Hui Sia

**Affiliations:** Department of Cardiology, National University Heart Centre, Singapore 119228, Singapore; ong.leong.tung@mohh.com.sg

**Keywords:** myocardial infarction, psychiatric disorders, anxiety disorder, major depressive disorder, bipolar disorder, schizophrenia, cardiovascular disease

## Abstract

This review explores the association between myocardial infarction (MI) and major psychiatric conditions, such as depression, bipolar disorder, and schizophrenia. Research shows that people with psychiatric illnesses are at a higher risk of developing heart disease, and vice versa. This relationship is influenced by factors such as chronic inflammation, stress hormone imbalances, genetic predisposition, and oxidative stress, which can damage both the heart and brain. For example, stress and inflammation in MI can lead to changes in brain chemicals that can contribute to depression, while oxidative damage in psychiatric conditions can contribute to heart disease. Additionally, genetic factors may play a role in both conditions, further strengthening their association. Understanding these shared mechanisms is essential for improving the treatment of individuals with both MI and psychiatric disorders. This review highlights the need for further research and better medical strategies to reduce the impact of these conditions and improve the quality of life for affected individuals.

## 1. Introduction

Myocardial infarction (MI) and psychiatric conditions are two major causes of disability and death worldwide [1]. Many studies have demonstrated that the survivors of MI are at an elevated risk of developing psychiatric disorders and vice versa [1]. However, the exact neurobiological pathways that link MI to psychiatric illness remain poorly understood [2]. Individuals with major psychiatric conditions have a life expectancy that is approximately 15 to 25 years shorter than that of the general population, with cardiovascular disease accounting for the majority of these premature deaths [3,4]. Cardiovascular disease is the leading cause of death among individuals with major psychiatric conditions with mortality rates being more than twice compared to the general population, and this has continued to rise over recent decades [5].

The relationship between myocardial infarction and cardiovascular disease appears to be bidirectional, with acute coronary events and chronic cardiovascular conditions potentially triggering the development of psychiatric conditions [6]. In addition, the evidence indicates that even among patients with low coronary artery calcium scores, the mortality rates in those with major psychiatric conditions remain three to four times higher than in the general population [7]. The emerging research highlights the shared pathophysiological mechanisms between myocardial infarction and psychiatric conditions, encompassing biological, neurohormonal and genetic factors [8]. This review aims to explore the molecular mechanisms between myocardial infarction and major psychiatric conditions.

## 2. Anxiety Disorder and Major Depressive Disorder

Figure 1 demonstrates the pathophysiology of myocardial infarction and depression. Studies have demonstrated that patients with MI have a higher risk of developing newly diagnosed anxiety and major depressive disorders [9]. MI serves as a significant risk factor for newly diagnosed clinical anxiety and depressive disorders within the first two years following the event [9]. The biopsychosocial model suggests that the interplay of biological, psychological, and social factors may contribute to the development of anxiety and depressive disorder post-myocardial infarction [10]. MI can lead to major depressive disorder facilitated by increased symptoms of psychological discomfort post-MI cardiac rehabilitation compared to the general population [11].

MI triggers responses like the hypothalamic–pituitary–adrenal (HPA) axis activation and autonomic nervous system (ANS) dysregulation, potentially leading to prefrontal cortex and anterior cingulate gyrus dysfunction, which contributes to depression and anxiety disorder [12]. In addition, MI activates the secretion of the corticotropin-releasing hormone (CRH) from the hypothalamus, stimulating the anterior pituitary gland to release the adrenocorticotropic hormone (ACTH) [2]. The ACTH stimulates the adrenal glands to produce cortisol, catecholamines, and an abnormal cortisol level leading to HPA axis dysfunction due to impaired feedback control. This study demonstrated that the cortisol levels in post-MI patients spike immediately after the event due to the HPA axis activation but return to normal within 72 h [13]. While post-MI patients with depression lasting over three months showed a flattened daily cortisol rhythm, those without depression had significantly lower afternoon cortisol levels compared to the morning [13]. Abnormal cortisol rhythms have been associated with cognitive impairment and diminished stress-coping abilities, potentially heightening the risk of developing major depressive disorder and anxiety disorder [13,14].

Inflammation is one of the main pathophysiological mechanisms in the development of depression in patients post-MI [15]. During MI, damage-associated molecular patterns (DAMPs), like heat shock proteins (HSPs) and high-mobility group box 1 (HMGB1), are released from the heart muscles due to oxygen deprivation [15]. These molecules activate the immune cells, such as the macrophages that release pro-inflammatory cytokines, including interleukin (IL)-6, IL-1β, and the tumor necrosis factor (TNF)-α, which is essential for the phagocytosis of damaged cardiac muscle and cardiac repair [15,16]. Inflammatory cytokines are implicated in the development of depression, as demonstrated in both animal and human studies [12]. Research in rodent models reveals that inflammatory cytokines, such as IL-1β and TNF-α, can induce depressive-like behavior, which is also observed in humans where elevated inflammatory markers may lead to a depressed mood [17,18]. Inflammatory cytokines can disrupt neurotransmitter systems, such as reducing serotonin levels via increased indoleamine 2,3-dioxygenase activity, and altering dopamine and norepinephrine metabolism and production leading to the dysregulation of mood [19,20,21]. In addition, inflammatory cytokines activate the kynurenine pathway, which increases quinolinic acid production, which activates the N-methyl-D-aspartate (NMDA) receptor leading to the development of depressive symptoms [22].

The plasminogen activator inhibitor 1 (PAI-1), encoded by the SERPINE1 gene, serves as a key inhibitor of the tissue plasminogen activator (tPA) in the extracellular space, and has been associated with an increased risk of depression and the response of selective serotonin reuptake inhibitors [23,24]. PAI-1 levels increase during periods of psychological stress and depression, while serotonergic antidepressant treatment has been associated with reduced PAI-1 levels [23,25]. Additionally, patients with depression often exhibit lower baseline tPA levels, which significantly increase following antidepressant therapy, highlighting a potential relationship between depression and MI [26]. Elevated PAI-1 and fibrinogen levels may inhibit fibrinolysis leading to the evaluated risk of MI [12]. The coagulation system may lead to the development of depression through the tPA-plasmin pathway, which converts the pro-Brain-derived neurotrophic factor (BDNF) into BDNF, a neurotrophin essential for synaptic plasticity and neuronal connectivity [27]. In addition, increased inflammatory cytokines have also been shown to decrease levels of BDNF [28,29]. Reduced BDNF levels, particularly in the brain regions involved in mood regulation, like the amygdala, prefrontal cortex, hippocampus, and amygdala, have been consistently linked to emotional stress and depression [30].

Mendelian randomization studies have demonstrated a significant link between a genetic predisposition to depression and a higher risk of developing MI [12]. Otte et al. identified an association between the serotonin transporter gene variant (5-HTTLPR) and an increased risk of depression in patients with myocardial infarction (MI), with carriers of this variant exhibiting a poorer response to treatment with antidepressants [31]. In addition, patients with both depression and MI have an increased sensitivity or upregulation of serotonin receptors and a reduced expression of the serotonin-transporter receptor leading to increased thromboembolic risk [32]. Furthermore, the S allele of the 5-HTT gene polymorphic region has been linked to both depressive symptoms and adverse cardiac outcomes [33]. The genetic differences in the IL-1 gene were demonstrated to have a higher likelihood of developing depression following an MI, potentially due to the increase in the inflammatory response during MI [34].

## 3. Bipolar Disorder

Figure 2 shows the pathophysiology of myocardial infarction and bipolar disorder. Patients with bipolar disorder experienced prolonged stress due acute mania or depression leading to a disrupted parasympathetic response [35]. The heart–brain axis allows communication between the cardiovascular and nervous systems, with sympathetic and parasympathetic activity mediated by acetylcholine, epinephrine, and norepinephrine to regulate cardiac contractility and heart rate variability [36,37]. Limited research has reported that patients with bipolar disorder have decreased heart rate variability, a key indicator of autonomic nervous system activity [38,39]. Reduced heart rate variability is associated with a 32–45% higher risk of the development cardiovascular disease [40]. During the acute phase of mania or depression in bipolar disorder, the HPA is activated, leading to the paraventricular nuclei to the secret CRH [35]. This stimulates the anterior pituitary gland to produce ACTH, leading to elevated cortisol levels [35]. Hypercortisolemia contributes to insulin resistance and hyperglycemia, leading to the increase in the release of pro-inflammatory cytokines [35]. The inflammatory cytokines damage the endothelial cells, accelerate the formation of the atherosclerotic plaques due to the oxidization of low-density lipoprotein (LDL), leading to cholesterol crystal build-up and MI [41].

The neurobiological basis of bipolar disorder may involve dysfunctions in the neurotrophic pathways and energy metabolism, with increased oxidative stress causing lipid and protein peroxidation, impairing signal transduction, structural plasticity, and cellular resilience [42]. In vivo magnetic resonance spectroscopy studies demonstrated altered levels of phosphocreatine, phosphomonoesters, and intracellular pH, highlighting the disruptions in oxidative phosphorylation, energy production, and phospholipid metabolism in bipolar disorders [42]. The acute phase of bipolar disorder leads to increased oxidative stress, characterized by elevated levels of oxidants, including peroxides and malondialdehyde, and reduced antioxidants, such as vitamin E, coenzyme Q10, and catalase that results in generating reactive oxygen species (ROS) through the pathways involving redox factor 1, activator protein 1, and the hypoxia-inducible factor 1 [43,44]. This oxidative damage contributes to endothelial wall injury, promoting atherosclerosis and cardiovascular diseases [45]. In addition, lipid peroxidation, driven by excess ROS damages polyunsaturated fatty acids in cell membranes, leading to the formation of lipid hydroperoxides and cellular dysfunction [46]. This oxidative damage can compromise mitochondrial integrity, disrupt energy metabolism, and trigger inflammatory responses, contributing to the development and progression of MI [46].

Mitochondrial dysfunction in bipolar disorder is influenced by increased oxidative stress, pro-inflammatory cytokines, and intracellular calcium levels, which are more pronounced during manic episodes [47]. Increased calcium ions and oxidative stress enhance oxidative phosphorylation through the pathways involving adenosine triphosphate (ATP) synthase, AMP-activated protein kinase (AMPK), SIRT-1, SIRT-3, and NAD+ [48]. Mitochondrial dysfunction in bipolar disorder, leading to excessive calcium and oxidative stress, may exacerbate myocardial injury in individuals with underlying ischemic heart disease. Excessive calcium entry into the mitochondria via the mitochondrial calcium uniporter (MCU) and impaired extrusion by the Na^+^/Ca^2^^+^ exchanger (NCX) results in mitochondrial calcium overload, leading to the opening of the mitochondrial permeability transition pore (MPTP) [49]. This, combined with oxidative stress-induced damage to mitochondrial membranes and proteins, results in ATP depletion, mitochondrial swelling, and cardiomyocyte death [50]. These processes exacerbate myocardial injury and infarct.

Bipolar disorder is associated with a deficiency of the methylenetetrahydrofolate reductase (MTHFR) enzyme, which is essential for homocysteine metabolism leading to increased central and peripheral homocysteine levels during the manic and euthymic phases [51]. The increased homocysteine levels lead to the NMDA receptor activation, causing increased intracellular calcium, ROS production, and subsequently cause neuronal autophagy or necrosis [52]. Hyperhomocysteinemia depletes the nitric oxide and antioxidants, leading to an increase in platelet-derived growth factors and the activation of clotting factors, which leads to platelet adhesion, aggregation, and thrombus formation [53]. In addition, hyperhomocysteinemia also disrupts the structural integrity of collagen, proteoglycans, and elastin, leading to arterial stiffening [54]. All these factors contribute to vessels occlusion and an increased risk of MI in patients with bipolar disorder [35].

## 4. Schizophrenia

Figure 3 shows the pathophysiology of myocardial infarction and schizophrenia. Inflammation and immune system dysfunction have been observed in individuals with schizophrenia [55]. Cytokines play an essential role in mediating the effects of inflammation in schizophrenia, potentially associating prenatal insults to the schizophrenia [55]. Prenatal infections and maternal immune system alterations have been identified as significant factors, increasing the likelihood of schizophrenia and related neurocognitive and neuroanatomical abnormalities in offspring [56]. Patients with psychosis may demonstrate a higher level of pro-inflammatory cytokines [57]. The activation of the inflammatory response system may lead to microglial activation, as evidenced by post-mortem studies showing an increase in the microglial density in the brains of patients with schizophrenia [58]. This process is believed to disrupt neuronal circuit development and function in the brain [59]. Cytokines may also induce neuronal apoptosis, potentially contributing to the functional brain deficits associated with schizophrenia [59]. Elevated levels of cytokines, such as IL-1β, IL-6, and the transforming growth factor-β, have been reported during acute relapses and the first episodes of schizophrenia but often normalize following antipsychotic treatment [60]. However other cytokines, including IL-12, IL-γ, TNF-α, and the soluble IL-2 receptor, remain elevated during acute relapses, first episodes, and even with antipsychotic therapy [38]. Increased levels of cytokines, including IL-1α, IL-6, IL-8, IL-12, IL-33, and IL-35, have a positive correlation with atherosclerosis and an increased risk of developing myocardial infarction [61].

Oxidative stress, characterized by an elevated reactive species and diminished antioxidant defenses, is implicated in cellular damage and has been consistently observed in schizophrenia [62]. The research assessing oxidative stress in schizophrenia has utilized several peripheral biomarkers, like reduced plasma, antioxidant and glutathione levels, and diminished antioxidant enzyme activities, including superoxide dismutase (SOD) and glutathione peroxidase [62]. Studies indicate that patients with acute coronary syndrome exhibit significantly lower SOD levels compared to healthy controls, with NSTEMI patients showing even lower levels than STEMI patients, suggesting more severe oxidative stress [63]. The reduced SOD and catalase levels may result from the increased utilization to neutralize the free radicals generated during myocardial ischemia, reflecting a depletion of antioxidant defenses [63]. In addition, approximately one-third of schizophrenia patients exhibit pronounced redox dysregulation, with acute-phase polyunsaturated fatty acid (PUFA) deficits and detrimental responses to eicosapentanoate (EPA) or vitamin E and C, although these effects stabilize during remission, marked by a persistent redox imbalance in low-PUFA subgroups [64,65]. A reduced serum PUFA level has been connected to an increased risk of cardiovascular events, including myocardial infarction [66].

Genome-wide association studies demonstrated the shared genetic variants associated with schizophrenia and cardiovascular diseases due to the similar pathophysiology of inflammation and metabolism [67]. Four distinct genetic loci (rs35044849, rs3118357, rs9257136, and rs9257248) have demonstrated significant colocalization between schizophrenia and cardiovascular disease, suggesting that these variants may influence both cardiovascular and neurological systems [68]. The shared genetic loci highlights a potential role for this locus in the heart–brain axis through the regulation of the CX3CL1 expression, which is a chemokine that is implicated in immune responses and neuroinflammation, potentially linking cardiovascular disease and schizophrenia [68]. CX3CL1 is overexpressed in atherosclerotic plaques, contributing to their formation and progression by recruiting inflammatory cells, like monocytes, to vascular walls leading to an increased risk of myocardial infarction [69]. In addition, elevated CX3CL1 levels have been observed in heart failure patients, correlating with an increased risk of cardiac dysfunction through the mechanisms involving inflammation and tissue remodeling [70]. Furthermore, the association of rs35044849 with various schizophrenia phenotypes and proHB-EGF reduces cardiac contractility, causes interstitial fibrosis and exacerbates cardiac remodeling after myocardial infarction leading to worsening cardiac function [68].

Substance use disorders, including tobacco, alcohol, cannabis, and cocaine use disorders, are frequently observed in patients with schizophrenia with a lifetime prevalence from 60% to 90% for cigarette smoking, 21% to 86% for alcohol use, and 17% to 83% for cannabis use [71]. Gene–environment interactions contribute to the risk of schizophrenia and co-occurring substance use disorders, with BDNF, catechol-O-methyltransferase (COMT), and protein kinase B (AKT) being the most studied genes linked to both conditions [71]. Furthermore, schizophrenia may stem from a dysfunction in the brain circuits related to reward and motivation, particularly the mesocorticolimbic dopamine system, leading to both substance initiation and continued use [71]. Substance use significantly contributes to an increased risk of MI, and a significant proportion of cases do not manifest any symptoms or signs of coronary artery disease, likely due to coronary microvascular dysfunction [72].

## 5. Future Research Directions

Major psychiatric disorders are linked to a reduced lifespan and signs of accelerated aging due to disruptions in circadian rhythms leading to oxidative stress, inflammation, and mitochondrial function [73]. Circadian disruptions are commonly observed in psychiatric conditions, often linked to reduced melatonin levels and the dysregulation of the HPA axis [74]. The suppression of pineal melatonin in the nighttime disrupts the regulation of the cortisol levels during sleep, affecting the morning cortisol awakening response, and the activated glucocorticoid receptor-alpha (GR-α) from moving into the nucleus [74]. The suppression of melatonin throughout the circadian cycle may disrupt the regulation of the HPA axis, affecting cardiovascular function and recovery after myocardial infarction [75]. Gut dysbiosis and the suppression of butyrate levels, which are typically evident in psychiatric conditions, reduce the butyrate’s ability to suppress the GR-α nuclear translocation from its cytoplasmic complex with Hsp-90 and p-23 [76]. Since pineal melatonin at night helps to seal the gut and prevent gut dysbiosis; therefore, its suppression may be closely linked to alterations in the gut microbiome [77]. Alterations to the gut microbiota increase the risk of myocardial infarction by modifying gut-produced metabolites and complex interplay with host genetics [78].

In addition, mood and psychotic disorders are often associated with hyperglycemia and metabolic dysregulation [79]. Hyperglycemia and hypertension increase methylglyoxal, which is typically modeled as mediating its effects through its role as a precursor for advanced glycation end-products (AGEs) and by activating the receptor for AGEs [80]. However, recent data indicate that methylglyoxal binds to and reduces the availability of tryptophan, thereby restricting the initiation of the tryptophan-serotonin-N-acetylserotonin-melatonin pathway [81]. As this pathway is also relevant to cardiovascular and cardiomyocyte function, the suppression of this pathway may, therefore, represent another overlapping pathophysiological feature linking mood/psychotic disorders with myocardial infarction [82].

## 6. Conclusions

This review highlights the bidirectional relationship between MI and major psychiatric conditions. The shared pathophysiological mechanisms, such as inflammation, oxidative stress, neurohormonal dysregulation, and genetic factors, underlie this connection between MI and psychiatric illnesses. Advancements in understanding the pathophysiological mechanisms provide critical insights regarding the complexities of managing patients with comorbid MI and psychiatric disorders. Further research is essential to identify the targeted interventions addressing these shared mechanisms to improve both mental and cardiovascular diseases. A holistic approach incorporating multidisciplinary care, early diagnosis, and innovative therapeutic strategies is essential for optimizing the outcomes in this vulnerable population.

## Figures and Tables

**Figure 1 biology-14-00336-f001:**
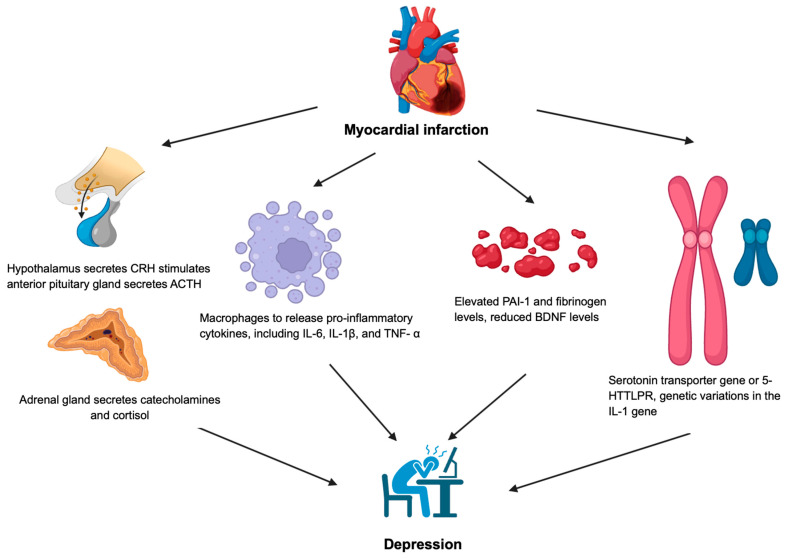
Pathophysiology of myocardial infarction and depression. ACTH: adrenocorticotropic hormone, BDNF: brain-derived neurotrophic factor, CRH: corticotropin-releasing hormone, IL: interleukin; PAI-1: plasminogen activator inhibitor 1, TNF-α: tumor necrosis factor-α; 5-HTTLPR: serotonin transporter gene variant.

**Figure 2 biology-14-00336-f002:**
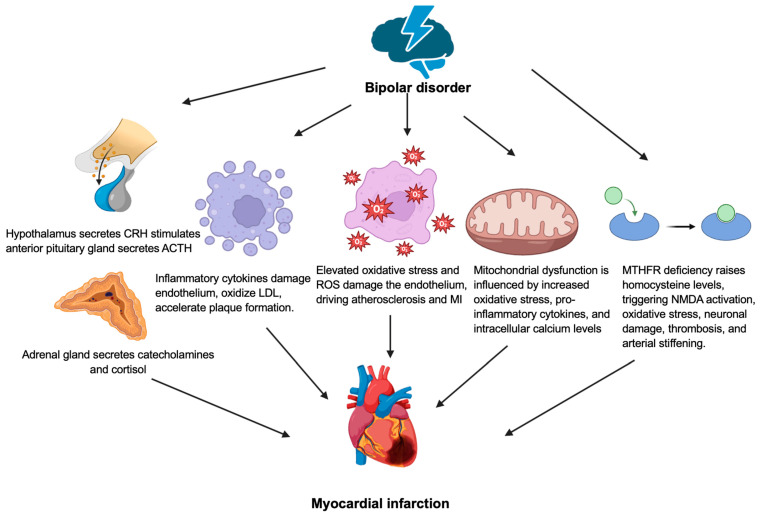
Pathophysiology of myocardial infarction and bipolar disorder. ACTH: adrenocorticotropic hormone, CRH: corticotropin-releasing hormone, LDL: low-density lipoprotein, MI: myocardial infarction, MTHFR: methylenetetrahydrofolate reductase, NMDA: N-methyl-D-aspartate, ROS: reactive oxygen species.

**Figure 3 biology-14-00336-f003:**
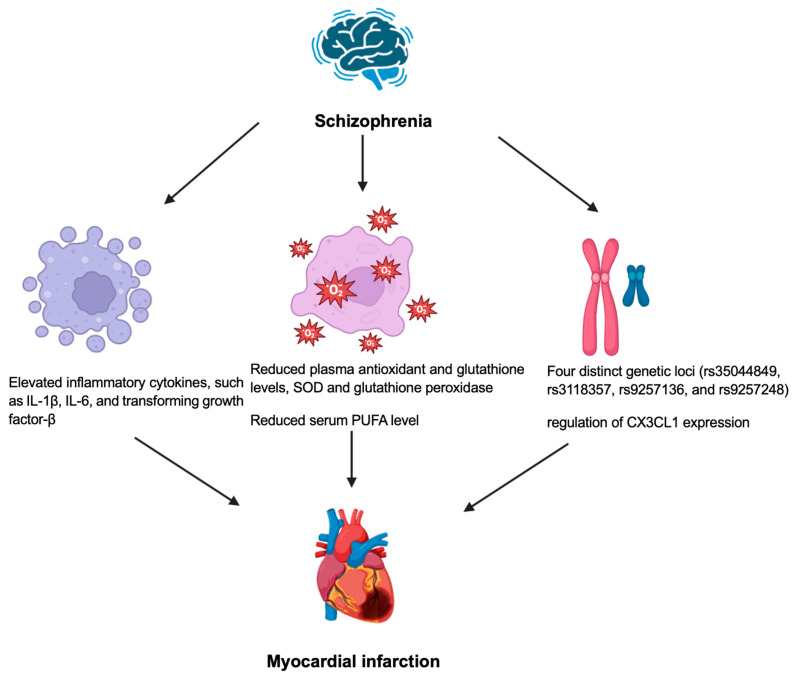
Pathophysiology of myocardial infarction and schizophrenia. IL: interleukin, PUFA: polyunsaturated fatty acid, SOD: superoxide dismutase.

## Data Availability

No new data were created or analyzed in this study.

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
