# Peer review of "Pathophysiological Links Between Myocardial Infarction and Anxiety Disorder, Major Depressive Disorder, Bipolar Disorder and Schizophrenia"

_biology, 2025, doi:10.3390/biology14040336_

Round 1
Reviewer 1 Report
Comments and Suggestions for Authors
Dear authors, I have been glad to revise this manuscript, as I think that the correlation between psychiatric conditions and heart disease can be often overlooked.
Please find suggestions in order to improve the ms:
1) in the paragraph concerning major depression the authors indicate the depression post IAM only. Actually, a comorbidity has been observed and the causal connection can be bidirectional.
Heart disease can be strictly connected to depression and may be facilitated by mood disorders and anxiety disorders. See, as an example, data on Takotsubo syndrome:
Ricci M, Pozzi G, Caraglia N, Chieffo DPR, Polese D, Galiuto L. Psychological Distress Affects Performance during Exercise-Based Cardiac Rehabilitation. Life (Basel). 2024 Feb 8;14(2):236. doi: 10.3390/life14020236. PMID: 38398745; PMCID: PMC10890595.
Mariano EG, Marconi M, Pozzi G, Locorotondo G, Cecchini E, Malci F, Sposini Ghezzi S, Polese D, Galiuto L. Psychosocial and psychopathological dimensions of patients with Takotsubo Syndrome. Panminerva Med. 2024 Dec;66(4):380-391. doi: 10.23736/S0031-0808.24.05187-5. Epub 2024 Dec 6. PMID: 39641305.
On the other hand the correlation with schizophrenia and bipolar disorders is better described. Nevertheless, the daily habits in these conditions are still overlooked. Smoke and a not healthy habits (i.e. food habits), which are caused by the pathological schizophrenic behavior can be implied into the pathogenetic mechanisms of psychopathology.
2) Please update the references about this content and also about the comorbidity between IAM and depression/mood disorders/ anxiety disorders.
Author Response
Comment: in the paragraph concerning major depression the authors indicate the depression post IAM only. Actually, a comorbidity has been observed and the causal connection can be bidirectional.
Heart disease can be strictly connected to depression and may be facilitated by mood disorders and anxiety disorders. See, as an example, data on Takotsubo syndrome:
Ricci M, Pozzi G, Caraglia N, Chieffo DPR, Polese D, Galiuto L. Psychological Distress Affects Performance during Exercise-Based Cardiac Rehabilitation. Life (Basel). 2024 Feb 8;14(2):236. doi: 10.3390/life14020236. PMID: 38398745; PMCID: PMC10890595.
Mariano EG, Marconi M, Pozzi G, Locorotondo G, Cecchini E, Malci F, Sposini Ghezzi S, Polese D, Galiuto L. Psychosocial and psychopathological dimensions of patients with Takotsubo Syndrome. Panminerva Med. 2024 Dec;66(4):380-391. doi: 10.23736/S0031-0808.24.05187-5. Epub 2024 Dec 6. PMID: 39641305.
On the other hand the correlation with schizophrenia and bipolar disorders is better described. Nevertheless, the daily habits in these conditions are still overlooked. Smoke and a not healthy habits (i.e. food habits), which are caused by the pathological schizophrenic behavior can be implied into the pathogenetic mechanisms of psychopathology.
Response: Thank you for your comment. We have included bidirectional relationship between depression and MI in the first paragraph of depression and last paragraph (genetic predisposition of depression leading to MI). We have included the relationship between schizophrenia, substance use disorder and MI.
2) Please update the references about this content and also about the comorbidity between IAM and depression/mood disorders/ anxiety disorders.
Response: Thank you for your comment. We have included bidirectional relationship between depression and MI in the first paragraph of depression and last paragraph (genetic predisposition of depression leading to MI). We have included the relationship between schizophrenia, substance use disorder and MI.
Reviewer 2 Report
Comments and Suggestions for Authors
Journal: Biology (ISSN 2079-7737)
Manuscript ID: biology-3490224
Type: Review
Title: Pathophysiological links between Myocardial Infarction and Anxiety Disorder, Major Depressive Disorder, Bipolar Disorder and Schizophrenia
Authors: Leong Tung Ong * , Ching-Hui Sia
Section: Neuroscience
This submission reviews the associations between major psychiatric disorders and myocardial infarction risk. The submission highlights shared risk factors, including genetic, autonomic nervous system and HPA axis, whilst emphasizing the roles of heightened oxidative stress and immune inflammatory factors. The manuscript is clearly organized and written.
Major
It would be useful for readers if something like the following was included before the Conclusions Section. The suggested additional content is meant as an example of more cutting-edge conceptualizations of relevant processes, which is clear and referenced and meant, as stated, as an example of what would be interesting to include as future research without requiring the manuscript to be re-written. Clearly, it does not have to be inserted verbatim.
e.g.
“As indicated above, major psychiatric conditions are associated with decreased longevity and indices of accelerated aging. Recent work indicates that accelerated aging may be associated with alterations in the circadian rhythm and how oxidative stress, inflammation and mitochondrial function are dampened and reset at night [Anderson, 2024]. Circadian disruption is evident in the psychiatric conditions reviewed above, typically in association with decreased melatonin and dysregulated HPA axis [Serrano and Serrano, 2021] and with implications for patient assessment and treatment [Colita et al., 2024]. The suppression of pineal melatonin at night results in a dysregulated influence of the rise in cortisol over sleep that peaks in the morning cortisol awakening response [Anderson, 2024previous]. Melatonin not only suppresses cortisol production by the adrenal cortex but also prevent the activated glucocorticoid receptor-alpha (GR-α) from being translocated to the nucleus [Anderson, 2023]. Any suppression of melatonin over the circadian rhythm may then act to dysregulate the influence of the HPA axis, including in the regulation of cardiovascular function and recovery after myocardial infarction [Sethi et al., 2023].
As with melatonin, gut dysbiosis and the suppression of butyrate levels typically evident in psychiatric conditions [Hadrich et al., 2025], attenuates the capacity of butyrate to suppress the GR-α nuclear translocation from its cytoplasmic complex with heat shock protein (hsp)90 and p-23 [Anderson, 2024previous]. As pineal melatonin at night seals the gut and prevents gut dysbiosis, the suppression of pineal melatonin may be intimately associated with alterations in the gut microbiome and its products to influence the psychiatric conditions highlighted [Zhang et al., 2022] as well as myocardial infarction [Yang et al., 2022].
Mood and psychotic disorders are often associated with hyperglycemia and metabolic dysregulation [Penninx et al., 2018]. Hyperglycemia and hypertension increase methylglyoxal, which is typically modelled as mediating its effects as a precursor for advanced glycation end products (AGEs) and the activation of the receptor for AGEs (RAGE). However, recent data indicates that methylglyoxal binds an limits the availability of tryptophan [Md Samsuzzaman et al., 2024], thereby limiting the initiation of the tryptophan-serotonin-N-acetylserotonin-melatonin pathway. As this pathway is also relevant to cardiovascular and cardiomyocyte function [Manjarrez-Gutiérrez et al., 2022], the suppression of this pathway may therefore be another overlapping pathophysiological feature of mood/psychotic disorder overlaps with myocardial infarction.”
This should provide readers with links to wider bodies of cutting-edge data pertinent to the overlaps of MI with mood/psychotic disorders, perhaps under a heading of ‘Future research directions’.
References
Anderson G. Physiological processes underpinning the ubiquitous benefits and interactions of melatonin, butyrate and green tea in neurodegenerative conditions. Melatonin Res. 2024;7(1):20-46. doi: 10.32794/mr11250016
Serrano-Serrano AB, Marquez-Arrico JE, Navarro JF, Martinez-Nicolas A, Adan A. Circadian Characteristics in Patients under Treatment for Substance Use Disorders and Severe Mental Illness (Schizophrenia, Major Depression and Bipolar Disorder). J Clin Med. 2021 Sep 25;10(19):4388. doi: 10.3390/jcm10194388. PMID: 34640406; PMCID: PMC8509477.
Colita CI, Hermann DM, Filfan M, Colita D, Doepnner TR, Tica O, Glavan D, Popa-Wagner A. Optimizing Chronotherapy in Psychiatric Care: The Impact of Circadian Rhythms on Medication Timing and Efficacy. Clocks Sleep. 2024 Nov 5;6(4):635-655. doi: 10.3390/clockssleep6040043. PMID: 39584972; PMCID: PMC11586979.
Anderson G. Melatonin, BAG-1 and cortisol circadian interactions in tumor pathogenesis and patterned immune responses. Explor Target Antitumor Ther. 2023;4(5):962-993. doi: 10.37349/etat.2023.00176. Epub 2023 Oct 25. PMID: 37970210; PMCID: PMC10645470.
Sethi Y, Padda I, Sebastian SA, Malhi A, Malhi G, Fulton M, Khehra N, Mahtani A, Parmar M, Johal G. Glucocorticoid Receptor Antagonism and Cardiomyocyte Regeneration Following Myocardial Infarction: A Systematic Review. Curr Probl Cardiol. 2023 Dec;48(12):101986. doi: 10.1016/j.cpcardiol.2023.101986. Epub 2023 Jul 20. PMID: 37481215.
Hadrich I, Turki M, Chaari I, Abdelmoula B, Gargouri R, Khemakhem N, Elatoui D, Abid F, Kammoun S, Rekik M, Aloulou S, Sehli M, Mrad AB, Neji S, Feiguin FM, Aloulou J, Abdelmoula NB, Sellami H. Gut mycobiome and neuropsychiatric disorders: insights and therapeutic potential. Front Cell Neurosci. 2025 Jan 8;18:1495224. doi: 10.3389/fncel.2024.1495224. PMID: 39845646; PMCID: PMC11750820.
Zhang B, Chen T, Cao M, Yuan C, Reiter RJ, Zhao Z, Zhao Y, Chen L, Fan W, Wang X, Zhou X, Li C. Gut Microbiota Dysbiosis Induced by Decreasing Endogenous Melatonin Mediates the Pathogenesis of Alzheimer's Disease and Obesity. Front Immunol. 2022 May 10;13:900132. doi: 10.3389/fimmu.2022.900132. PMID: 35619714; PMCID: PMC9127079.
Yang L, Wang T, Zhang X, Zhang H, Yan N, Zhang G, Yan R, Li Y, Yu J, He J, Jia S, Wang H. Exosomes derived from human placental mesenchymal stem cells ameliorate myocardial infarction via anti-inflammation and restoring gut dysbiosis. BMC Cardiovasc Disord. 2022 Feb 17;22(1):61. doi: 10.1186/s12872-022-02508-w. PMID: 35172728; PMCID: PMC8851843.
Penninx BWJH, Lange SMM. Metabolic syndrome in psychiatric patients: overview, mechanisms, and implications. Dialogues Clin Neurosci. 2018 Mar;20(1):63-73. doi: 10.31887/DCNS.2018.20.1/bpenninx. PMID: 29946213; PMCID: PMC6016046.
Md Samsuzzaman, Hong SM, Lee JH, Park H, Chang KA, Kim HB, Park MG, Eo H, Oh MS, Kim SY. Depression like-behavior and memory loss induced by methylglyoxal is associated with tryptophan depletion and oxidative stress: a new in vivo model of neurodegeneration. Biol Res. 2024 Nov 21;57(1):87. doi: 10.1186/s40659-024-00572-4. PMID: 39574138; PMCID: PMC11580208.
Manjarrez-Gutiérrez G, Valero-Elizondo G, Serrano-Hernández Y, Mondragón-Herrera JA, Mansilla-Olivares A. Hypertrophic cardiomyopathy induces changes in the tryptophan-5-hydroxylase, serotonin transporter and serotonergic receptors expressions. Gac Med Mex. 2022;158(6):386-392. English. doi: 10.24875/GMM.M22000717. PMID: 36657118.
Author Response
Comment 1:
This submission reviews the associations between major psychiatric disorders and myocardial infarction risk. The submission highlights shared risk factors, including genetic, autonomic nervous system and HPA axis, whilst emphasizing the roles of heightened oxidative stress and immune inflammatory factors. The manuscript is clearly organized and written.
Major
It would be useful for readers if something like the following was included before the Conclusions Section. The suggested additional content is meant as an example of more cutting-edge conceptualizations of relevant processes, which is clear and referenced and meant, as stated, as an example of what would be interesting to include as future research without requiring the manuscript to be re-written. Clearly, it does not have to be inserted verbatim.
e.g.
“As indicated above, major psychiatric conditions are associated with decreased longevity and indices of accelerated aging. Recent work indicates that accelerated aging may be associated with alterations in the circadian rhythm and how oxidative stress, inflammation and mitochondrial function are dampened and reset at night [Anderson, 2024]. Circadian disruption is evident in the psychiatric conditions reviewed above, typically in association with decreased melatonin and dysregulated HPA axis [Serrano and Serrano, 2021] and with implications for patient assessment and treatment [Colita et al., 2024]. The suppression of pineal melatonin at night results in a dysregulated influence of the rise in cortisol over sleep that peaks in the morning cortisol awakening response [Anderson, 2024previous]. Melatonin not only suppresses cortisol production by the adrenal cortex but also prevent the activated glucocorticoid receptor-alpha (GR-α) from being translocated to the nucleus [Anderson, 2023]. Any suppression of melatonin over the circadian rhythm may then act to dysregulate the influence of the HPA axis, including in the regulation of cardiovascular function and recovery after myocardial infarction [Sethi et al., 2023].
As with melatonin, gut dysbiosis and the suppression of butyrate levels typically evident in psychiatric conditions [Hadrich et al., 2025], attenuates the capacity of butyrate to suppress the GR-α nuclear translocation from its cytoplasmic complex with heat shock protein (hsp)90 and p-23 [Anderson, 2024previous]. As pineal melatonin at night seals the gut and prevents gut dysbiosis, the suppression of pineal melatonin may be intimately associated with alterations in the gut microbiome and its products to influence the psychiatric conditions highlighted [Zhang et al., 2022] as well as myocardial infarction [Yang et al., 2022].
Mood and psychotic disorders are often associated with hyperglycemia and metabolic dysregulation [Penninx et al., 2018]. Hyperglycemia and hypertension increase methylglyoxal, which is typically modelled as mediating its effects as a precursor for advanced glycation end products (AGEs) and the activation of the receptor for AGEs (RAGE). However, recent data indicates that methylglyoxal binds an limits the availability of tryptophan [Md Samsuzzaman et al., 2024], thereby limiting the initiation of the tryptophan-serotonin-N-acetylserotonin-melatonin pathway. As this pathway is also relevant to cardiovascular and cardiomyocyte function [Manjarrez-Gutiérrez et al., 2022], the suppression of this pathway may therefore be another overlapping pathophysiological feature of mood/psychotic disorder overlaps with myocardial infarction.”
This should provide readers with links to wider bodies of cutting-edge data pertinent to the overlaps of MI with mood/psychotic disorders, perhaps under a heading of ‘Future research directions’.
References
Anderson G. Physiological processes underpinning the ubiquitous benefits and interactions of melatonin, butyrate and green tea in neurodegenerative conditions. Melatonin Res. 2024;7(1):20-46. doi: 10.32794/mr11250016
Serrano-Serrano AB, Marquez-Arrico JE, Navarro JF, Martinez-Nicolas A, Adan A. Circadian Characteristics in Patients under Treatment for Substance Use Disorders and Severe Mental Illness (Schizophrenia, Major Depression and Bipolar Disorder). J Clin Med. 2021 Sep 25;10(19):4388. doi: 10.3390/jcm10194388. PMID: 34640406; PMCID: PMC8509477.
Colita CI, Hermann DM, Filfan M, Colita D, Doepnner TR, Tica O, Glavan D, Popa-Wagner A. Optimizing Chronotherapy in Psychiatric Care: The Impact of Circadian Rhythms on Medication Timing and Efficacy. Clocks Sleep. 2024 Nov 5;6(4):635-655. doi: 10.3390/clockssleep6040043. PMID: 39584972; PMCID: PMC11586979.
Anderson G. Melatonin, BAG-1 and cortisol circadian interactions in tumor pathogenesis and patterned immune responses. Explor Target Antitumor Ther. 2023;4(5):962-993. doi: 10.37349/etat.2023.00176. Epub 2023 Oct 25. PMID: 37970210; PMCID: PMC10645470.
Sethi Y, Padda I, Sebastian SA, Malhi A, Malhi G, Fulton M, Khehra N, Mahtani A, Parmar M, Johal G. Glucocorticoid Receptor Antagonism and Cardiomyocyte Regeneration Following Myocardial Infarction: A Systematic Review. Curr Probl Cardiol. 2023 Dec;48(12):101986. doi: 10.1016/j.cpcardiol.2023.101986. Epub 2023 Jul 20. PMID: 37481215.
Hadrich I, Turki M, Chaari I, Abdelmoula B, Gargouri R, Khemakhem N, Elatoui D, Abid F, Kammoun S, Rekik M, Aloulou S, Sehli M, Mrad AB, Neji S, Feiguin FM, Aloulou J, Abdelmoula NB, Sellami H. Gut mycobiome and neuropsychiatric disorders: insights and therapeutic potential. Front Cell Neurosci. 2025 Jan 8;18:1495224. doi: 10.3389/fncel.2024.1495224. PMID: 39845646; PMCID: PMC11750820.
Zhang B, Chen T, Cao M, Yuan C, Reiter RJ, Zhao Z, Zhao Y, Chen L, Fan W, Wang X, Zhou X, Li C. Gut Microbiota Dysbiosis Induced by Decreasing Endogenous Melatonin Mediates the Pathogenesis of Alzheimer's Disease and Obesity. Front Immunol. 2022 May 10;13:900132. doi: 10.3389/fimmu.2022.900132. PMID: 35619714; PMCID: PMC9127079.
Yang L, Wang T, Zhang X, Zhang H, Yan N, Zhang G, Yan R, Li Y, Yu J, He J, Jia S, Wang H. Exosomes derived from human placental mesenchymal stem cells ameliorate myocardial infarction via anti-inflammation and restoring gut dysbiosis. BMC Cardiovasc Disord. 2022 Feb 17;22(1):61. doi: 10.1186/s12872-022-02508-w. PMID: 35172728; PMCID: PMC8851843.
Penninx BWJH, Lange SMM. Metabolic syndrome in psychiatric patients: overview, mechanisms, and implications. Dialogues Clin Neurosci. 2018 Mar;20(1):63-73. doi: 10.31887/DCNS.2018.20.1/bpenninx. PMID: 29946213; PMCID: PMC6016046.
Md Samsuzzaman, Hong SM, Lee JH, Park H, Chang KA, Kim HB, Park MG, Eo H, Oh MS, Kim SY. Depression like-behavior and memory loss induced by methylglyoxal is associated with tryptophan depletion and oxidative stress: a new in vivo model of neurodegeneration. Biol Res. 2024 Nov 21;57(1):87. doi: 10.1186/s40659-024-00572-4. PMID: 39574138; PMCID: PMC11580208.
Manjarrez-Gutiérrez G, Valero-Elizondo G, Serrano-Hernández Y, Mondragón-Herrera JA, Mansilla-Olivares A. Hypertrophic cardiomyopathy induces changes in the tryptophan-5-hydroxylase, serotonin transporter and serotonergic receptors expressions. Gac Med Mex. 2022;158(6):386-392. English. doi: 10.24875/GMM.M22000717. PMID: 36657118.
Response: Thank you for your suggestion. We have included the section above in the future research directions section.
Round 2
Reviewer 2 Report
Comments and Suggestions for Authors
The suggested changes have been incorporated and the manuscript is now suitable for publication.